# New Growth Curves for Spanish Children (0–10 Years) in the Region of Extremadura

**DOI:** 10.3390/ijerph18178953

**Published:** 2021-08-25

**Authors:** Luis Pardo-Galán, Raquel Pastor-Cisneros, Daniel Collado-Mateo, José Carmelo Adsuar, Miguel Ángel García-Gordillo, Lucía Bautista-Bárcena

**Affiliations:** 1Promoting a Healthy Society Research Group (PHeSO), Faculty of Sport Sciences, University of Extremadura, 10003 Cáceres, Spain; lpardog@alumnos.unex.es (L.P.-G.); jadssal@unex.es (J.C.A.); 2Centre for Sport Studies, Rey Juan Carlos University, 28943 Fuenlabrada, Spain; daniel.collado@urjc.es; 3Universidad Autónoma de Chile, Sede Talca, Maule 3467987, Chile; miguelgarciagordillo@gmail.com; 4Mathematics Department, Faculty of Sport Sciences, Polytechnic School of Caceres, University of Extremadura, 10003 Cáceres, Spain; luciabb@unex.es

**Keywords:** children’s health, growth, LMS method, percentile, sex, size, weight

## Abstract

The anthropometric reference data used to examine the growth pattern of children in Spain are obtained from studies carried out several years ago. In the region of Extremadura, the tables obtained by the Faustino Obergozo Foundation, which date back to 2004, are used. The first objective of this work is to develop growth tables and graphs that accurately reflect the somatometric variables of children in Extremadura. Secondly, the averages of these variables by sex will be compared to determine if there are significant differences between them. A database provided by the General Directorate of Planning, Training, and Health and Social Quality of the Regional Government of Extremadura was used, which contains the measurements of height, weight, and body mass index (BMI) of boys and girls in Extremadura between the years 2006–2016. The database was analyzed using the Statistical Package for the Social Sciences (SPSS) version 23 and the R software version 3.5.1, considering a cross-sectional study. As a result, the tables and growth graphs of Extremadura’s population base for weight and height are presented, from birth to 10 years, as well as comparisons of the average values of the analyzed variables between boys and girls. We found that there are significant differences in the mean values, according to sex, of the height and weight. On the other hand, BMI progressed normally when comparing the results of the Extremadura population with those reflected by the World Health Organization (WHO). Differences were found when comparing the results with those obtained by the Faustino Orbegozo Foundation.

## 1. Introduction

The study of human growth is defined as the process by which individuals increase their mass and height as they reach maturity, acquiring the functional characteristics of the adult state. To understand the development of the species, it provides fundamental data on the state of health of the whole and the society in which it develops [1].

This process is limited by genetic factors, and these, in turn, are influenced by extra-genetic factors that determine the rate of maturity and the final height as the result of a complex interaction between both.

Growth is one of the characteristic physiological processes of childhood. It is usually monitored by the pediatrician because it is considered a health indicator. Changes in anthropometric variables such as weight, height, and body mass index (BMI) in children have been demonstrated in different long-term studies on growth trends [2]. The growth trends of generations are also indicative of the health status of a particular population. Height has historically been considered an indicator of the health and living standards of human populations and is used as a benchmark to determine differences between populations [3].

Growth charts are an essential tool for monitoring childhood and adolescent development and are commonly used to diagnose and monitor changes in height, body weight, and BMI [2]. Similarly, these references to the state of growth are very useful when assessing the well-being of individuals as well as other aspects indicative of social equality. It is widely known that improvements in nutritional and socio-health aspects lead to an increase in the growth rate of the population in which they are applied [4].

Therefore, in order to benefit from such profits, an analysis of growth patterns and a comparison of the individuals’ data with a set of standardized values obtained from a representative sample of the population to which the individual belongs are required. This is the objective of cross-sectional design studies, which allow these parameters to be determined by comparing their results with those obtained in longitudinal design studies.

The lack of determining standards in Spain has led to different investigations developed in different places such as in the city of Bilbao [5], the Autonomous Community of Murcia [6], or in Andalusia [7]. Many of these studies are used in different populations and are based on data collected in one or several centers, generally following a heterogeneous data collection methodology. Specifically, in the Autonomous Community of Extremadura, the tables obtained by the Faustino Obergozo Foundation, last updated in 2004, are used [8]. From 2004 to the present day, it is possible that the growth pattern of the children has been modified. On the other hand, it is also possible that the economic conditions of the place where a child resides affect his/her later growth.

The fact that the monitoring of child and adolescent growth in Extremadura is being carried out with tables based on anthropometric data from children living in the Basque Country may lead to an erroneous generalization of data. Furthermore, it is important to bear in mind that differences in the development index (HDI) have been found between the most developed territories in Spain, such as the Basque Country, and other less developed ones, such as Extremadura [9]. Therefore, the development and growth of young people in various territories could present important differences in adolescents taking into account the particular life context of each region.

One of the shortcomings of this type of study is the variability in the methodologies used to collect locally collected data, generalizing the results to populations with very different characteristics from those found in the original studies. For this reason, the aim of this work is to develop tables and graphs in the Autonomous Community of Extremadura for the weight and height of children from birth to 10 years of age.

The approach to the creation of these growth tables and graphs has sought to refine the original database using the two-step transformation method, seeking to obtain values of normality that would allow the performance of the Student T-test to compare independent means.

### 1.1. Objectives and Hypotheses

The data were collected from various pediatrics offices in health centers and private practices in Extremadura. The data were collected in Pediatrics offices in health centers, primary care centers, and hospitals in the community of Extremadura, both public and private. The eligibility criteria were children registered in that community and born between 2006 and 2016. This database was provided by the General Directorate of Planning, Training, and Health and Social Quality of the Regional Government of Extremadura.

The main objectives of the work are:Develop growth tables that reflect the somatometric variables of children in Extremadura.Develop graphs that reflect the somatometric variables of children in Extremadura.Compare the average of these variables by sex.Determine if there are significant differences between boys and girls.

We start with the following hypothesis:Children born in different parts of the autonomous community of Extremadura between 2006 and 2016 were selected.It is assumed that we have a sufficiently representative sample of subjects to correctly elaborate the tables and growth curves. For this purpose, we used a convenience sample with a power of 95%.The weight and height of the subjects was obtained by their Primary Health Care Center.The normality of the variables studied is verified. If this is not the case, we must correct the asymmetry of the variable by the Two-Step Transformation Procedure described below, with the help of the statistical software SPSS.

### 1.2. Methodology

In this section, the construction of percentile tables from the LMS method is explained.

Studies aimed at producing percentile tables traditionally use the LMS method (Lambda, Mu, Sigma), which will be explained briefly below [10].

Normally, anthropometric variables do not follow a normal distribution since they present different degrees of asymmetry in even kurtosis. The LMS method is widely used to correct the asymmetry of these variables. This method does not correct the kurtosis, so it is used to calculate the percentiles of an anthropometric variable when the histogram of the frequencies is like the normal one, but it presents some very accentuated tails that affect its symmetry.

The LMS methodology was adopted by the International Obesity Task Force (IOTF) to develop growth curves for children and adolescents [10].

One of the most outstanding advantages of this methodology is the existence of explicit formulas for the calculation of any desired percentile and the possibility of comparing the results with other studies. The disadvantage is that these methods show a certain sensitivity to extreme outliers, such as, for example, data from children with obesity. This manifests itself with large values of the estimated degrees of freedom, which generates complex and not very smooth percentile curves. One solution may be to remove extreme outliers from the sample. Another drawback of LMS methods is that the curves need to be updated at least once in a decade.

The application of the LMS method consists of a transformation known as parameterization, which calculates the Z-scores needed to calculate the percentiles as long as the kurtosis is 0. It is called the LMS method because of the appearance of three parameters (Lambda (*λ*), Mu (*μ*), and Sigma (*σ*)) in the transformation equation.
(1)$$z_{LMS}=\frac{1}{\sigma_{L}\lambda }\left[{\left(\frac{y}{\mu }\right)}^{\lambda }-1\right]\;\;\;for\;\;y,\mu ,\sigma_{L}\;and\;\;\lambda \neq 0$$

If the variable available presents kurtosis, it will be necessary to resort to another process such as Box–Cox potency exponential (BCPE), where an additional parameter (tau) is added to correct the kurtosis.

The application of these two transformations (Lambda-Mu-Sigma or Box–Cox power exponential) can be carried out with specific statistical software such as R. In this work, these LMS methods have been used to calculate the percentile tables of the weight variable. Once this graph was obtained, it was compared with a graph obtained by calculating the sample percentiles.

## 2. Materials and Methods

### 2.1. Study Design

A descriptive and correlational study was carried out on the records obtained from the database provided by the Directorate General for Planning, Training, and Health and Social Health Quality of the Regional Government of Extremadura and collected from different Primary Care centers.

### 2.2. Participants

Participants included 58,586 individuals from birth to 10 years of age. Of the total sample, 30,463 (52%) were male and 28,123 (48%) females.

### 2.3. Measures and Procedures

To produce these tables, and for the corresponding comparisons between groups, a database provided by the General Directorate of Planning, Training, and Health and Social Health Quality of the Regional Government of Extremadura was used. This database was created in the Primary Care Pediatrics consultations in the Community of Extremadura and included the measurements of certain anthropometric variables of children in Extremadura (sex, date of birth, date of data collection, weight, height, and cranial perimeter) from 2006 to 2016. For the purposes of the tables, the child’s age is calculated from the difference between the date of data collection and the child’s date of birth, expressed in months. The database was analyzed using SPSS version 23 (IBM Corp., Armonk, NY, USA). Since a comparison between groups is to be made and the additional assumption of normality is required, if any of the variables analyzed do not follow a normal distribution, a transformation is made so that the assumption of normality is fulfilled.

The procedure for measuring a child’s height and weight is necessary to accurately calculate the BMI. Before attempting the procedure, we should check if the subject is able to stand upright. If this is not possible, the height is obtained using a metric stadiometer. A precision scale is used to calculate the child’s weight.

A study of the records obtained from the database provided by the General Directorate of Planning, Training, and Health and Socio-sanitary Quality of the Regional Government of Extremadura, collected from different primary care centers, was carried out. After eliminating outliers, the resulting database consisted of 131,238 records, obtained from 58,586 subjects, of which 30,463 (52%) are male and 28,123 (48%) female. The subjects in the resulting database corresponded to children born between 13 October 2006 and 12 September 2016. The health centers in which the data was taken correspond to 487 different populations in the autonomous community of Extremadura.

In our study, there were no variables that acted as confounding factors. Confounding exists when the association between two variables differs significantly depending on whether or not another variable is considered. This occurs when the association between two variables varies according to the different levels of one or more other variables. When calculating the weight and height of a child, there may be other variables that have an influence, but essentially the main one is age, which is the reason for our study. Therefore, there were no other sources of variability: we calculated the standard deviation of each of the variables. A cross-sectional study was carried out to assess the situation of a child at a specific moment in time, comparing it with the general population of its age and sex. The table includes the standard deviation, which is essential for assessing children at the extremes of the growth curves so that subjects of different ages can be compared with each other. A cross-sectional study has the advantage of being a quick study that shows the situation of an individual at a given time in relation to the reference population. However, it does not provide growth rates, as in the case of the longitudinal studies, where the children are followed until the end of growth. The longitudinal studies have some drawbacks: they are too long and there may be biases due to sample loss or due to relevant socio-economic changes during that period. For that reason, sample selection in growth studies is key. It must be representative of the reference population, with different age and sex groups. Anthropometric data should be obtained accurately, using accurate and properly calibrated instruments.

### 2.4. Statistical Analysis

The data were analyzed with the statistical software SPSS version 23. The comparison of means between different anthropometric variables in males and females was carried out using the Student *t*-test, considering that there was a significant difference in the means of the two groups (males and females) for a given variable when the *p*-value of the Student *t* contrast was less than 0.05.

To stratify the variable by age, the newborn group was all those records with an age equal to 0. For the 3-month-old group, those aged between 0 and 3 months (the latter included). For the 6-month-old group, those aged between 3 and 6 months (the latter included). This same criterion was repeated successively in each of the subsequent subgroups.

In each of the age intervals, for both men and women, the percentiles (3, 10, 25, 50.75, 90, and 97) of weights and heights were estimated. These percentiles are the most referenced in the specialized literature [5,8,11].

To apply the Student T to contrast and compare the mean values of each anthropometric variable between men and women, two assumptions needed to be verified: the assumption of normality and assumption of homoscedasticity. The assumption of normality implies that the variables within each group follow a normal distribution. The assumption of homoscedasticity implies that the variances of the variables within each group are equal.

The normality of the anthropometric variables for each age group was checked and it was found that, on numerous occasions, this hypothesis of normality was not fulfilled. An example of this statement can be found in Figure 1, where the asymmetry of the variable weight can be seen, and due to this asymmetry, the variable weight does not present a normal distribution. To correct the asymmetry of the variable, the so-called “process of transformation into two steps” was applied to continuous variables [12].

The “two-step transformation process” was employed using SPSS version 23 using the following procedure.

Assign a fractional range to the variable and generate a new variable with this range assignment.Calculation of a new variable. This is acheived by selecting it in the function group panel GL Inverse and within this category “Idf. Normal”. This function requires three parameters: the variable generated in step 1, the mean, and standard deviation of the original variable.

Figure 2 shows the histogram corrected by this two-step transformation procedure and the correction of the asymmetry of the variable is visually observed.

Table 1 and Table 2 show the *p*-values of the Kolmogorov–Smirnov contrast normality for the original variables and the *p*-value of the Kolmogorov–Smirnov contrast normality for the variable corrected by the two-step transformation method. As can be seen, those groups in which normality is not achieved considerably improve the asymmetry and kurtosis of the treated variable, for example, the weight of children up to 36 months, where the kurtosis goes from 3.82 to −0.048 and the asymmetry from 1.208 to 0.008. In a normal distribution, these coefficients of asymmetry and kurtosis are equal to zero).

In cases where normality is not verified (such as in the weight group of children up to 36 months), it is not possible to apply the Student *t*-test. However, when applying this two-step transformation method, the variable is symmetrical. In a symmetric distribution, mean and median coincide so that when a non-parametric test is used, specifically the Mann–Whitney U-test to compare the medians between both groups, the mean between groups of the variables is compared.

#### Construction of a Percentile Table for the Variable Weight Using the GAMLSS Package in R

The construction of the percentile chart of the weight variable will now be explained. The R software was used to create this percentile graph—specifically the GAMLSS Generalized Additive Models for Location, Scale, and Shape package. This package is a general framework for fitting regression models where the location, scale, and shape of the response variable vary according to the values of the explanatory variable [13,14]. In addition, the World Health Organization recommends the use of the GAMLSS package for the construction of these percentile tables [15]. Figure 3 shows these percentile charts obtained with the R software (Ross Ihaka & Robert Gentleman, Auckland, New Zealand).

## 3. Results

Table 3 and Table 4 show the results of the comparison of averages of the weight and height variables according to age and sex.

Table 5 shows the average and standard deviation of the BMI of the sample.

Table 6 shows the sample percentiles 3, 10, 25.50 (median), 75, 90, and 97 of the weight and height for each sex, respectively, according to age. In contrast, Figure 3 shows the weight percentile table using R’s GAMLSS package. The representation of these percentiles is shown in Figure 4 and Figure 5, and the comparison of these percentiles with the ones obtained by the Faustino Orbegozo Foundation [8] is shown in Figure 6 and Figure 7. Finally, Figure 8 shows the average BMI of the sample.

Table 3, Table 4 and Table 5 show the differences between the sexes with respect to weight, height, and BMI. In the case of height, there are significant differences between boys and girls up to the age of 7 years, including the latter. In the case of weight, there are significant differences between boys and girls up to the age of 8 years, including the last one and except for 6 years of age. In the case of BMI, there are significant differences between boys and girls at birth, in the first three months of life, and in the first and second year.

## 4. Discussion

The set of subjects studied and data represented in the tables of this work constitute the most important and recent volume of somatometric data attributed to children in the autonomous community of Extremadura. These new graphs can be very useful in the medical health field, as they allow a more precise classification of the infant according to their height and weight for their age and graphic location.

The results obtained show a significant difference between the average growth rates of boys and girls from birth to over 8 years of age. This reinforces the previous knowledge about sexual dimorphism from very early ages [11]. At the same time, they solve a problem that has been repeatedly discussed in the specialized literature: the inconvenience of extrapolating the data obtained to other populations and the need for each community to have its own reference values. In this sense, there are different studies in our country which try to establish the patterns of normal growth in different areas [7,8,16,17].

The present study was compared with similar data obtained from studies in other autonomous communities [17]. Regarding differences between boys and girls, no major differences were found between the groups of Extremadura and in other places. It is only noteworthy that both weight and height values in girls remain slightly below those of boys, especially at early ages. In adolescence, the tendency is to equalize between the two groups.

It is shown that in 2010, regional differences in growth in Spain have disappeared and, in addition, adult height has approached that of other European and American countries, although it is still below that of some northern European countries [17]. In this study, children from Andalusia, Barcelona, Bilbao, and Zaragoza were considered. The results are practically the same as those obtained in our work. In this study, the importance given to the early detection of childhood overweight and obesity should be emphasized since it is observed that their prevalence has increased in the last two decades.

To analyze obesity, which generally affects the health of our country and is especially incipient at an early age, the average BMI in the different age groups was calculated. As can be seen in Figure 8, BMI between boys and girls was very similar, especially between 48 and 120 months. In the first months of life, they begin to be practically equal (between 3 and 6 months). However, in the period between 6 and 36 months, girls had a lower BMI than boys. This is reflected in comparative Table 5. In the first stage of life, the mean differs more between boys and girls, although it should be noted that the standard deviation was lower than after 36 months, when this standard deviation begins to increase, indicating greater variability at older ages.

We can perform a more exhaustive analysis of the results obtained by comparing them with the data of the Orbegozo Faustino Obergozo Foundation and the WHO. The percentile curves follow a similar evolution in boys and girls from birth to the age of 120 months. It is important to highlight that the first is a national study and the other is an international study. When comparing the percentiles that define overweight and obesity, it is observed that the BMI values obtained in our work are higher, although the evolution of the curves is similar. In girls, something similar to that observed in boys is observed. At lower ages, no differences are observed in the evolution of BMI. However, differences are observed in the higher ages, where the values from the Extremadura study are higher than those derived from the Orbegozo study, and the differences are greater when the age is higher. Regarding the WHO data, there are no differences. Comparing the values of BMI evolution in girls according to age, our study shows that the values are quite similar to the WHO study at any age.

The BMI can be considered a socio-economic indicator of the area in which it is used, i.e., it is a marker of the balance or imbalance of the economic level between different areas. It should be noted that the community of Extremadura has different characteristics from the rest of the Spanish communities. It is an isolated and rural region, which lags somewhat behind the more developed areas in terms of infrastructure and technological advances. However, as has been noted, no significant differences were found. 

This paper compares the results obtained with those of the study carried out by the Faustino Obergonzo Eizaguirre Foundation in the Basque Country in 2000 [8]. The aim was to observe the differences between the two, and to see that they follow a similar development, with the most pronounced differences appearing in the weight of the individuals, in both sexes. These differences may be due to the disparity between populations, as some studies point out [18], or to limitations in the data collection.

One of the limitations of this work is the precision and reliability of the results according to the measuring instruments used by the personnel in charge of carrying out the measurements at the Health Center.

Under experimental conditions, the homogeneity of the instruments and the reduction of staff responsible for carrying out the measurements could significantly improve the results. Despite this, there was heterogeneity in the materials and personnel between different communities, which could more accurately reflect this work rather than what happens in daily practice. The results could also be used more generally. For these reasons, this limitation can also be interpreted as a strength.

Another weakness of this paper that should be highlighted is the variability in the number of records for everyone. As can be seen from the results, the number of records varies significantly between age subgroups. It is individuals with pathology or peculiarity who appear in the greatest number of records (due to the medical follow-up carried out), compared to completely healthy and normative individuals, who go for more regular reviews over a greater time period.

These weaknesses may be the cause of the excessive “noise” or dispersion that we can observe in the distribution of the data when comparing Figure 3 (obtained by the SML method) and Figure 4, Figure 5, Figure 6, Figure 7 and Figure 8 (obtained using the data resulting from the two-step transformation). In the latter, we can observe an increasing dispersion from the age of 60 months onwards for both sexes. This is because, from the age of 5, children no longer require such a systematic review of their anthropometric variables by what revisions are distanced in time and the children who are monitored on a continuous basis. In this regard, the WHO recommends that the scope of surveys related to anthropometric indicators should extend up to 60 months of age [19].

Due to the limited scope of this work, the geographical locations of the individuals in the sample have not been differentiated. This information was collected in the original database. It allows certain growth patterns to be established according to geographical distribution or other characteristics and population, such as ethnicity or socioeconomic level [20]—this study is planned for a later project.

The mainly descriptive nature of the study means that relationships, such as cause-effect, cannot be established. The experimental treatment of the data is quite simple, as it is a descriptive observational study. This may be one of the main limitations of the work.

However, it is useful and has external validity, as it is used to determine if there is a delay in the child’s development (for example, if the child changes percentile in a short time), and it is also a tool for monitoring and following up the child’s normal growth.

As for the selection and size of the sample, the sample size was not calculated because it was a convenience sample. However, the power of the sample for the study has been considered, placing it at 95%.

The eligibility criteria were the children of the health centers of Extremadura who were healthy and between these ages, using the JARA program of the Junta de Extremadura.

To conclude, it is important to highlight the novelty of this article, as there are no tables or graphs, specific by sex and age, that represent the growth of children in Extremadura. The reference tables obtained corroborated the significant differences between boys and girls, mainly up to 8 years of age with respect to the variables studied; except for 6 years of age in the case of weight and 8 years of age for height. The tables and graphs presented in this work could be useful as normative and reference data to improve the classification evaluation of the growth of children in the autonomous community of Extremadura. At the same time, they could be functional for the detection and evaluation of pathological or irregular growth processes or for the evaluation of temporary intergenerational growth trends in the future.

Studies have shown that improved maternal health, coupled with nutritional factors and healthier living habits, progressively increase the height of newborns. Therefore, the regular updating of growth data, as well as the specificity of these data, is highly recommended in order to know and assess the rate of population growth in a given region.

## Figures and Tables

**Figure 1 ijerph-18-08953-f001:**
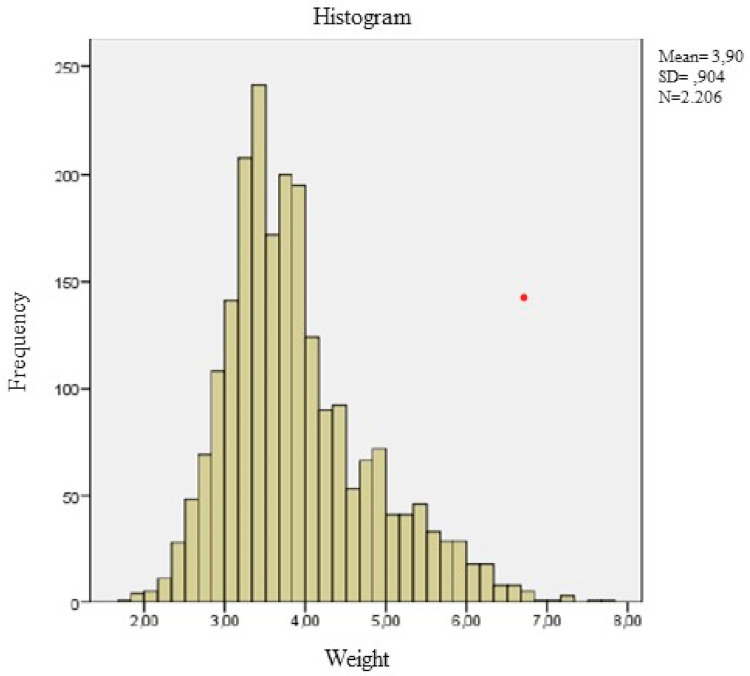
Distribution of the initial variable weight.

**Figure 2 ijerph-18-08953-f002:**
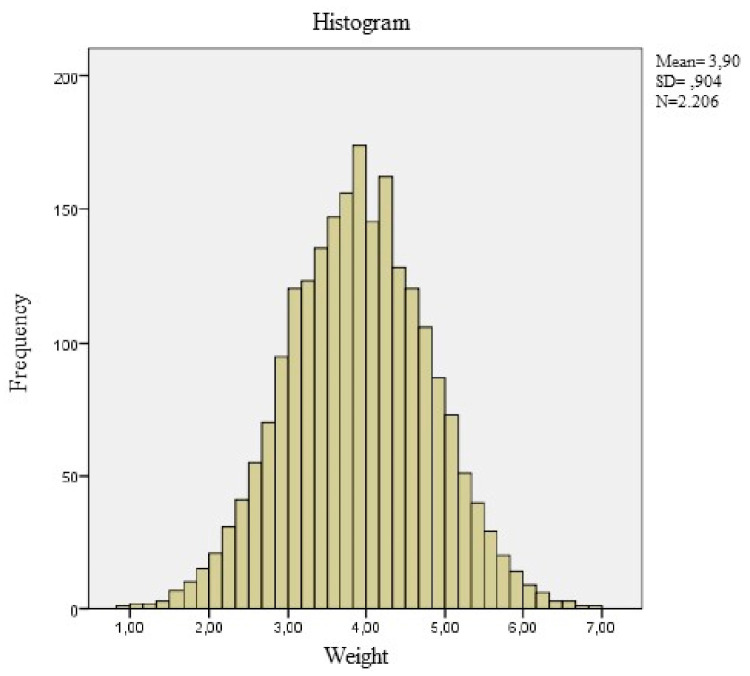
Distribution of the variable the two-step transformation process.

**Figure 3 ijerph-18-08953-f003:**
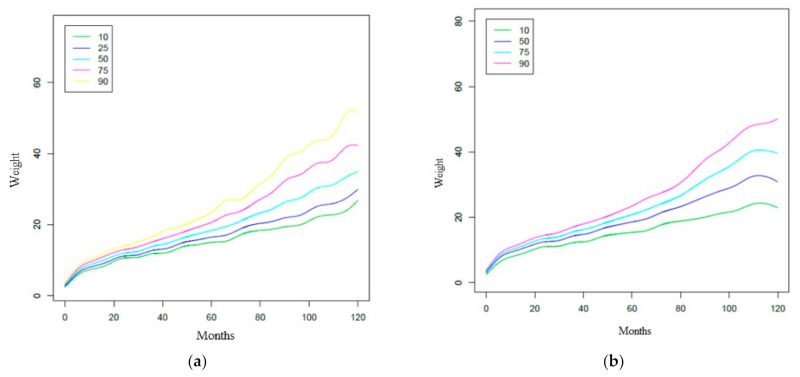
Percentile weight curves calculated with LMS method: (**a**) percentile weight curves in women; (**b**) percentile weight curves in men.

**Figure 4 ijerph-18-08953-f004:**
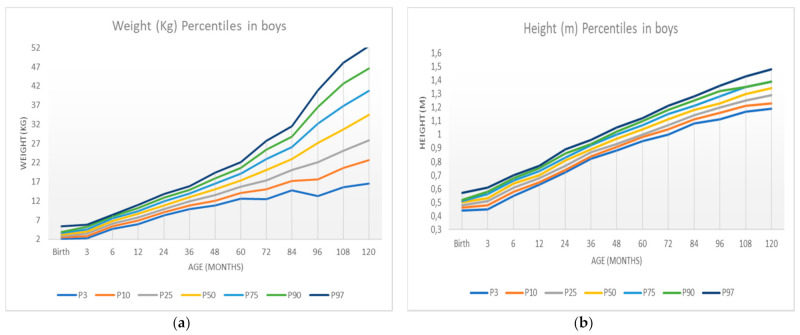
Weight and height percentiles in boys according to age: (**a**) weight percentiles; (**b**) height percentiles.

**Figure 5 ijerph-18-08953-f005:**
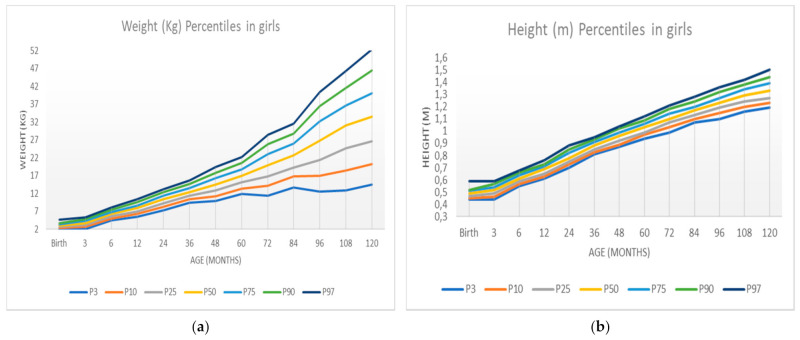
Weight and height percentiles in girls according to age: (**a**) weight percentiles; (**b**) height percentiles.

**Figure 6 ijerph-18-08953-f006:**
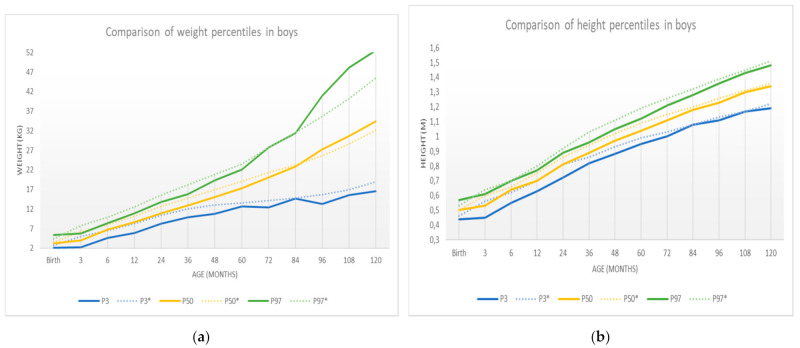
New specific curves for boys (solid) compared to the curves in [8] (dashed): (**a**) weight-specific curves; (**b**) height-specific curves.

**Figure 7 ijerph-18-08953-f007:**
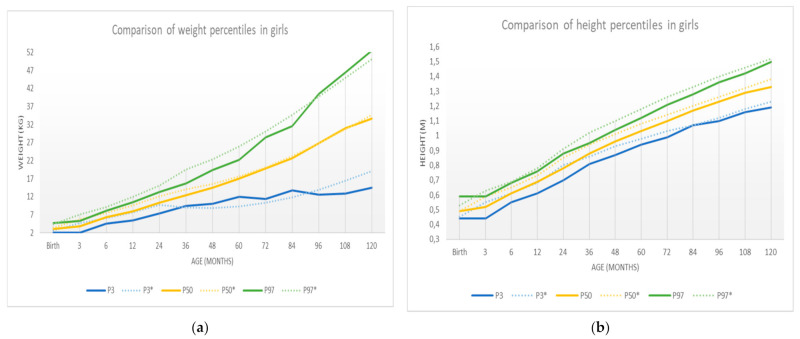
New specific curves for girls (solid) compared to the curves in [8] (dashed): (**a**) weight-specific curves; (**b**) height-specific curves.

**Figure 8 ijerph-18-08953-f008:**
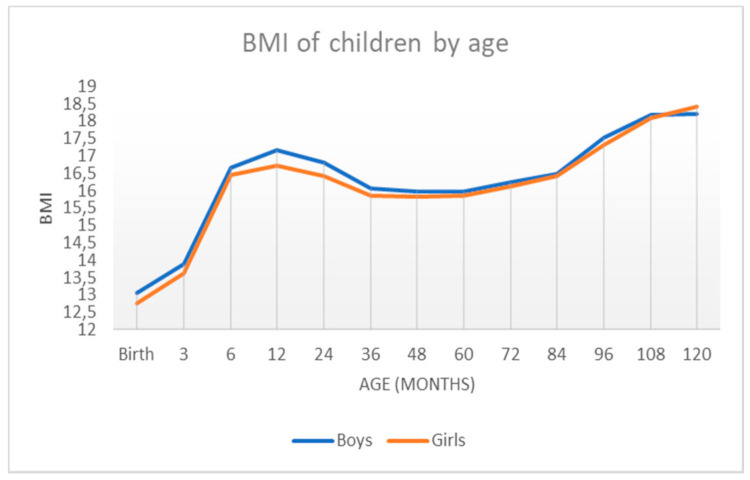
BMI in boys and girls according to age.

**Table 1 ijerph-18-08953-t001:** Differences in kurtosis, asymmetry of weight and height from birth to 3 years of age after the two-step transformation process.

Variables	Differences	Birth		3		6		12		24		36	
		Boys	Girl	Boys	Girls	Boys	Girls	Boys	Girls	Boys	Girls	Boys	Girls
Original Weight	*p*-value	0.200	0.191	0.000	0.000	0.043	0.060	0.000	0.009	0.000	0.100	0.000	0.000
	Kurtosis	NA	NA	0.588	0.728	NA	NA	8.248	NA	0.520	NA	3.829	4.298
	Asymmetry	NA	NA	0.892	0.834	NA	NA	0.040	NA	0.602	NA	1.209	1.388
Original Height	*p*-value	0.000	0.000	0.000	0.000	0.000	0.000	0.000	0.000	0.000	0.000	0.000	0.000
	Kurtosis	NA	NA	7.53	−0.79	1.016	−0.258	0.333	3.299	3.865	3.299	1.549	1.628
	Asymmetry	NA	NA	1.23	−0.837	−0.577	−1.358	0.143	−0.461	−0.167	−0.350	0.499	0.556
Treaty Weight	*p*-value	NA	NA	0.200	NA	NA	NA	0.200	NA	0.009	NA	0.000	0.000
	Kurtosis	NA	NA	−0.77	−0.79	−0.175	−0.258	−0.164	−0.076	−0.104	−0.127	−0.009	−0.013
	Asymmetry	NA	NA	0.011	0.001	NA	NA	−0.143	NA	0.163	NA	0.008	0.020
Treaty Height	*p*-value	0.000	0.000	0.000	0.000	0.000	0.006	0.005	0.013	0.000	0.001	0.000	0.000
	Kurtosis	NA	NA	−0.77	−0.79	−0.175	−0.258	−0.164	−0.076	−0.104	−0.127	−0.009	−0.013
	Asymmetry	NA	NA	0.006	0.005	−0.020	−0.035	0.002	0.055	0.000	−0.001	0.007	0.007

NA: not available; *p*-value: significant differences.

**Table 2 ijerph-18-08953-t002:** Differences in kurtosis, asymmetry of weight and height from 4 to 10 years old after the two-step transformation Process.

Variables	Differences	48		60		72		84		96		108		120	
		Boys	Girl	Boys	Girls	Boys	Girls	Boys	Girls	Boys	Girls	Boys	Girls	Boys	Girl
Original Weight	*p*-value	0.000	0.000	0.000	0.000	0.000	0.000	0.000	0.000	0.000	0.000	0.000	0.000	0.000	0.000
	Kurtosis	5.505	6.309	2.440	2.161	2.502	1.810	2.315	2.202	2.039	0.821	1.525	0.281	1.487	1.933
	Asymmetry	1.460	1.622	1.078	1.065	1.366	1.296	1.289	1.254	1.319	1.030	1.245	0.829	1.125	1.243
Original Height	*p*-value	0.000	0.000	0.000	0.000	0.000	0.000	0.000	0.000	0.000	0.000	0.000	0.001	0.000	0.000
	Kurtosis	1.871	2.792	0.789	1.182	0.200	0.056	0.871	2.059	−0.055	−0.138	0.595	−0.038	24.693	31.104
	Asymmetry	0.493	0.672	0.130	0.296	0.134	0.088	0.239	0.401	0.009	0.107	0.059	0.001	−2.273	−2.616
Treaty Weight	*p*-value	0.000	0.000	0.000	0.000	0.004	0.017	0.000	0.000	0.200	0.200	0.200	0.200	0.200	0.200
	Kurtosis	−0.002	−0.025	−0.001	−0.002	−0.045	−0.046	−0.014	−0.017	−0.056	−0.058	−0.078	−0.015	−0.127	−0.128
	Asymmetry	0.007	0.000	0.003	0.004	0.000	0.000	0.001	0.000	0.001	0.000	0.001	0.027	0.002	−0.001
Treaty Height	*p*-value	0.000	0.000	0.000	0.000	0.000	0.000	0.000	0.000	0.000	0.000	0.001	0.005	0.015	0.200
	Kurtosis	0.004	−0.019	0.006	−0.003	−0.005	−0.010	−0.012	−0.012	−0.062	−0.063	−0.077	0.014	−0.121	−0.123
	Asymmetry	0.010	0.004	0.04	0.003	0.013	0.015	0.001	0.001	0.000	0.002	0.000	0.027	−0.001	−0.001

*p*-value: significant differences.

**Table 3 ijerph-18-08953-t003:** Height (m) comparisons between boys and girls by age.

Height					
Age	Boys		Girls		*p*
	*n*	Mean ± SD	*n*	Mean ± SD	
Birth	441	0.49 ± 0.27	376	0.48 ± 0.25	<0.001
3 Months	1176	0.53 ± 0.03	1031	0.52 ± 0.38	<0.001
6 Months	316	0.63 ± 0.03	234	0.61 ± 0.31	<0.000
12 Months	397	0.70 ± 0.03	335	0.68 ± 0.04	<0.000
24 Months	807	0.80 ± 0.44	627	0.79 ± 0.04	<0.000
36 Months	11,267	0.89 ± 0.37	9930	0.88 ± 0.38	<0.000
48 Months	8148	0.96 ± 0.04	7253	0.95± 0.04	<0.000
60 Months	22,461	1.04 ± 0.04	20,561	1.03 ± 0.04	0.000
72 Months	3682	1.11 ± 0.05	3189	1.10 ± 0.05	0.000
84 Months	15,593	1.17 ± 0.05	14,415	1.17 ± 0.05	0.000
96 Months	2235	1.24 ± 0.06	2130	1.23 ± 0.06	0.000
108 Months	1338	1.29 ± 0.07	1275	1.29 ± 0.07	0.005
120 Months	656	1.34 ± 0.07	624	1.33 ± 0.08	0.159

**Table 4 ijerph-18-08953-t004:** Weight (kg) comparisons between boys and girls by age.

Weight					
Age	Boys		Girls		*p*
	*n*	Mean ± SD	*n*	Mean ± SD	
Birth	441	3.2 ± 0.55	376	3.05 ± 0.25	0.000
3 Months	1176	4.02 ± 0.92	1031	3.76 ± 0.86	0.000
6 Months	316	6.71 ± 0.98	234	6.31 ± 0.95	0.000
12 Months	397	8.54 ± 1.27	335	7.92 ± 1.28	0.000
24 Months	807	10.94 ± 1.44	627	10.33 ± 1.54	0.000
36 Months	11,267	12.90 ± 1.61	9930	12.48 ± 1.59	0.000
48 Months	8148	15.00 ± 2.27	7253	14.65 ± 2.43	0.000
60 Months	22,461	17.40 ± 2.52	20,561	17.00 ± 2.72	0.000
72 Months	3682	20.23 ± 4.09	3189	19.91 ± 4.50	0.255
84 Months	15,593	23.06 ± 4.50	14,415	22.75 ± 4.66	0.000
96 Months	2235	27.37 ± 7.02	2130	26.72 ± 7.54	0.003
108 Months	1338	31.12 ± 8.65	1275	30.53 ± 8.82	0.084
120 Months	656	34.31 ± 9.27	624	33.67 ± 9.94	0.240

**Table 5 ijerph-18-08953-t005:** BMI comparisons between boys and girls by age.

BMI					
Age	Boys		Girls		*p*
	*n*	Mean ± SD	*n*	Mean ± SD	
Birth	441	13.04 ± 1.42	376	12.75 ± 1.39	0.003
3 Months	1176	13.89 ± 1.86	1031	13.61 ± 1.94	0.001
6 Months	316	16.66 ± 1.72	234	16.46 ± 1.75	0.179
12 Months	397	17.15 ± 1.81	335	16.73 ± 2.00	0.003
24 Months	807	16.79 ± 1.56	627	16.43 ± 1.72	0.000
36 Months	11,267	16.06 ± 1.41	9930	15.85 ± 1.55	0.077
48 Months	8148	15.96 ± 1.69	7253	15.81 ± 1.91	0.005
60 Months	22,461	15.96 ± 1.65	20,561	15.84 ± 1.84	0.042
72 Months	3682	16.24 ± 2.36	3189	16.13 ± 2.69	0.057
84 Months	15,593	16.49 ± 2.39	14,415	16.43 ± 2.66	0.042
96 Months	2235	17.52 ± 3.49	2130	17.31 ± 3.86	0.090
108 Months	1338	18.18 ± 4.04	1275	18.09 ± 3.71	0.560
120 Months	656	18.21 ± 4.03	624	18.41 ± 4.51	0.089

**Table 6 ijerph-18-08953-t006:** Weight and height percentiles in boys and girls according to age.

Age (Months)	Variables	P3		P10		P25		P50		P75		P90		P97	
		Boys	Girls	Boys	Girls	Boys	Girls	Boys	Girls	Boys	Girls	Boys	Girls	Boys	Girls
Birth	Weight	2.14	2.05	2.50	2.32	2.86	2.66	3.24	3.08	3.59	3.42	3.93	3.73	5.43	4.72
	Height	0.44	0.44	0.46	0.45	0.48	0.47	0.50	0.49	0.51	0.51	0.52	3.73	0.57	0.59
3	Weight	2.28	2.08	2.81	2.67	3.42	3.16	4.02	3.75	4.61	4.37	5.22	4.87	5.80	5.32
	Height	0.45	0.44	0.48	0.46	0.51	0.49	0.53	0.52	0.56	0.54	0.58	0.57	0.61	0.59
6	Weight	4.67	4.53	5.43	5.07	6.09	5.73	6.71	6.33	7.49	6.89	7.91	7.46	8.39	8.17
	Height	0.55	0.55	0.58	0.57	0.61	0.59	0.64	0.61	0.66	0.64	0.68	0.66	0.70	0.68
12	Weight	5.92	5.49	6.89	6.25	7.70	7.03	8.59	7.92	9.38	8.67	10.13	9.59	10.96	10.44
	Height	0.63	0.61	0.65	0.63	0.68	0.65	0.70	0.69	0.73	0.71	0.75	0.73	0.77	0.76
24	Weight	8.28	7.37	9.12	8.31	9.92	9.23	10.89	10.34	11.90	11.45	12.86	12.33	13.83	13.25
	Height	0.72	0.70	0.74	0.73	0.77	0.75	0.81	0.78	0.83	0.82	0.86	0.85	0.89	0.88
36	Weight	9.92	9.41	10.86	10.39	11.90	11.33	12.95	12.42	14.00	13.53	14.94	14.61	15.86	15.61
	Height	0.82	0.81	0.84	0.83	0.87	0.85	0.89	0.88	0.92	0.91	0.93	0.93	0.96	0.95
48	Weight	10.82	10.01	12.09	11.32	13.49	12.80	15.07	14.53	16.47	16.27	17.92	17.83	19.36	19.36
	Height	0.88	0.87	0.91	0.89	0.93	0.92	0.97	0.96	1.00	0.99	1.02	1.02	1.05	1.04
60	Weight	12.66	11.92	14.11	13.40	15.67	15.11	17.35	16.97	19.06	18.83	20.60	20.53	22.12	22.23
	Height	0.95	0.94	0.98	0.97	1.00	0.99	1.04	1.03	1.07	1.06	1.10	1.09	1.12	1.12
72	Weight	12.41	11.44	15.05	14.24	17.30	16.88	20.10	19.89	22.95	23.08	25.43	25.90	27.73	28.43
	Height	1.00	0.99	1.04	1.03	1.07	1.07	1.11	1.10	1.15	1.14	1.18	1.18	1.21	1.21
84	Weight	14.71	13.75	17.17	16.80	20.03	19.34	22.92	22.66	26.00	26.00	28.77	28.76	31.47	31.54
	Height	1.08	1.07	1.11	1.10	1.14	1.13	1.18	1.17	1.21	1.20	1.25	1.24	1.28,	1.28
96	Weight	13.34	12.62	17.64	16.92	22.11	21.44	27.21	26.84	32.17	32.14	36.54	36.53	40.95	40.41
	Height	1.11	1.10	1.16	1.15	1.20	1.19	1.23	1.23	1.28	1.27	1.32	1.32	1.36	1.36
108	Weight	15.60	12.92	20.60	18.44	25.16	24.70	30.71	31.03	36.83	36.62	42.66	41.60	48.15	46.35
	Height	1.17	1.16	1.21	1.20	1.25	1.24	1.30	1.29	1.35	1.34	1.35	1.38	1.43	1.42
120	Weight	16.55	14.46	22.69	20.32	27.78	26.66	34.46	33.57	40.79	40.15	46.65	46.48	52.48	52.45
	Height	1.19	1.19	1.23	1.23	1.29	1.27	1.34	1.33	1.39	1.39	1.39	1.44	1.48	1.50

## Data Availability

The datasets used during the current study are available from the corresponding author on reasonable request.

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
