# Peer review of "New Growth Curves for Spanish Children (0–10 Years) in the Region of Extremadura"

_ijerph, 2021, doi:10.3390/ijerph18178953_

Round 1

Reviewer 1 Report

Using the database from authorities of Extremadura, this submission is going to refresh the growth pattern of children established in 2004. I like to give the following comments.

  1. Please update the data between 2010 – 2020 to reach the real application. Data before 2016 will be old soon.
  2. Limitation(s) and/or disadvantage(s) of LMS methods were not introduced.
  3. Age of the used participants was not indicated in clear.
  4. In Figure 1 or 2, treatment means what? Same as that in Table 1 or 2. Please indicate the treatment in legends or methods in clear.
  5. In Table 1 or 2, p value means the difference with what? It remained unknown.
  6. Difference between boys and girls seems not compared in clear, either weight and height.
  7. Changes in BMI were close to others or not. Please discuss it in detail.
  8. From the age of 5, as indicated before conclusion, children no longer require such a systematic review of their anthropometric variables. Please add the reference(s) to support. Additionally, limitation(s) and/or weakness need more.
  9. Conclusion did not show in clear, such as the variations between boys and girls for weight or height. Additionally, novelty of current report is important.

Author Response

Thank you for your comments. Please see attached file

Reviewer 2 Report

STRUCTURE

  • The manuscript is properly structured

TITLE AND ABSTRACT

  • The title or abstract should inform that the type of study
  • The grammar of the abstract needs to be revised, it is difficult to understand

INTRODUCTION

  • Line 48: add reference
  • Explain the scientific background in more depth. More background literature is needed to support the research
  • Section 1.1. corresponds to the methodology
  • State specific objectives and hypotheses

MATERIAL AND METHODS

Study design

  • Present key elements of study design in more depth

Setting

  • Describe the settings and locations where the data were collected

Participants

  • Give the eligibility criteria, and the sources and methods of selection of participants

Variables

  • Define clearly all exposures, predictors, potential confounders, and effect modifiers

Bias

  • Describe any efforts to address potential sources of bias

Study size

  • How sample size was determined?

RESULTS

  • In-depth explanation of the results shown in the tables and figures is lacking
  • This section should be reworded to ensure proper understanding of the data

DISCUSSION

  • There is a lack of discussion with other research and studies with similar objectives or where the importance of this research is highlighted
  • Give a cautious overall interpretation of results considering objectives, limitations, multiplicity of analyses, results from similar studies, and other relevant evidence
  • Discuss the generalisability (external validity) of the study results
  • The conclusion is weak and unclear

REFERENCES

  • The way of referencing in the text should be checked, some of them are not correct, for example line 264
  • References follow the indicated style

Author Response

(The authors gave the same response as above.)

Reviewer 3 Report

Title: should be representative of the data presented, current form is not accurate; for example: New growth curves for Spanish children (0-10 yrs) in the region of Extremadura

L41: madurity?

L43-44: ‘’ It usually be controlled by the paedatrician’’; the sentence needs rewriting

L253: use children or subjects instead of patients

Major issue: add details for methods of measuring BW and BH since these parameters are the foundation of the study

Author Response

(The authors gave the same response as above.)

Round 2

Reviewer 1 Report

It has been revised according to suggestions.

Reviewer 2 Report

No further comments. 

Reviewer 3 Report

Thank you for revising the manuscript.